# Common misconceptions held by health researchers when interpreting linear regression assumptions, a cross-sectional study

**Lee Jones**[1,2,3]*, **Adrian Barnett**[2], **Dimitrios Vagenas**[1]

**1** Research Methods Group, Faculty of Health, School of Public Health and Social Work, Queensland University of Technology, Kelvin Grove, Queensland, Australia, **2** AusHSI, Centre for Healthcare Transformation, Faculty of Health, School of Public Health and Social Work, Queensland University of Technology, Kelvin Grove, Queensland, Australia, **3** Statistics Unit, QIMR Berghofer Medical Research Institute, Herston, Queensland, Australia

* lee.jones@qut.edu.au

**Data availability statement:** The raw data and a reproducible R Quarto file used to produce this

## Abstract

**Background**: Statistical models are valuable tools for interpreting complex relationships within health systems. These models rely on a framework of statistical assumptions that, when correctly addressed, enable valid inferences and conclusions. However, failure to appropriately address these assumptions can lead to flawed analyses, resulting in misleading conclusions and contributing to the adoption of ineffective or harmful treatments and poorer health outcomes. This study examines researchers' understanding of the widely used linear regression model, focusing on assumptions, common misconceptions, and recommendations for improving research practices.

**Methods**: One hundred papers were randomly sampled from the journal *PLOS ONE*, which used linear regression in the materials and methods section and were from the health and biomedical field in 2019. Two independent volunteer statisticians rated each paper for the reporting of linear regression assumptions. The prevalence of assumptions reported by authors was described using frequencies, percentages, and 95% confidence intervals. The agreement of statistical raters was assessed using Gwet's statistic.

**Results**: Of the 95 papers that met the inclusion and exclusion criteria, only 37% reported checking any linear regression assumptions, 22% reported checking one assumption, and no authors checked all assumptions. The biggest misconception was that the Y variable should be checked for normality, with only 5 of the 28 papers correctly checking the residuals for normality.

**Conclusion**: The reporting of linear regression assumptions is alarmingly low. When assumptions are checked, the reporting is often inadequate or incorrectly checked. Addressing these issues requires a cultural shift in research practices, including improved statistical training, more rigorous journal review processes, and a broader understanding of regression as a unifying framework. Greater emphasis must be placed on evaluating model assumptions and their implications rather than the rote application of statistical methods. Careful consideration of assumptions helps improve the reliability of

paper, including all tables and figures have been stored in a GitHub repository https://github.com/Lee-V-Jones/Reporting_Linear_Regression_Assumptions and can be accessed at https://zenodo.org/records/10645770.

**Funding:** There was no cost associated with this research except for attending conferences. These costs were covered by the primary author's PhD allocation from the health faculty, Queensland University of Technology, and scholarships. The Statistical Society of Australia (SSA) and the Association for Interdisciplinary Meta-research & Open Science (AIMOS) supported the primary author with travel grants to attend their respective conferences. These scholarships did not influence the results of the study. The funders had no role in study design, data collection and analysis, decision to publish, or preparation of the manuscript.

**Competing interests:** The authors have declared that no competing interests exist.

statistical conclusions, reducing the risk of misleading findings influencing clinical practice and potentially affecting patient outcomes.

## Introduction

Medical research relies on the ability of researchers to verify and build on previous work. Researchers are continuously publishing new findings that can be used to develop new treatments for diseases and inform public policy. Dissemination of research through publication in peer-reviewed journals is a critical step in the scientific process that requires rigorous methods to be applied to ensure treatments are effective and appropriate [1]. Evaluation and improvement of research practices are essential to identifying flawed studies and improving the quality and reproducibility of research, a core focus of meta-research that investigates biases throughout the research process [2,3].

Statistical models provide tools to understand relationships in health systems by exploring data variability, estimating the effectiveness of new treatments and gaining better understanding of disease pathways. Unfortunately, when statistical methods are used poorly, they can provide misleading results, leading to wasted resources and patients receiving ineffective or even harmful treatments [4,5]. The underlying statistical assumptions should be satisfied for statistical tests to be reliable. If assumptions of tests are not met, the results may be misleading. At best, this may cause estimates to be inaccurate without changing the study's conclusion. At worst, assumption violations can cause results to be invalid, with the original findings found to be incorrect. Discussion of statistical assumptions is frequently absent from publications [6], with one study in the biomedical area showing assumptions were mentioned in only 20% of papers [7].

Poor statistical practice and reporting are pervasive across many disciplines [6,8,9], including ecology [10] and the social sciences [11]. With King et al. [12] identifying a research-to-practice gap, where applied researchers are often called upon to use statistical methods without sufficient expertise [13,14]. Arguably Ronald Fisher, one of the most influential statisticians of the 20th century, opened the doors to applied researchers with the publication of *Statistical Methods for Research Workers* in 1925, enabling the practical use of statistics across many fields [15]. However, it is unlikely Fisher could have envisioned the future of accessible statistical programs where users do not require technical understanding to produce results.

The consequences of misapplied statistical methods extend far beyond academic research. While the COVID-19 pandemic exposed widespread issues in the application of statistical methods, leading to misleading claims about treatments and interventions [16], a more persistent concern lies in the everyday medications that millions rely on. A recent example is the efficacy of oral phenylephrine as a nasal decongestant [17,18]. Despite being approved and widely available in over-the-counter cold medications, systematic reviews and regulatory assessments have found that oral phenylephrine is no more effective than a placebo [18]. This raises questions about how statistical weaknesses in past trials allowed it to remain on the market for so long and highlights the challenges of withdrawing ineffective medications once they are approved. These issues emphasise the need for statistically rigorous trials, as weak evidence at the approval stage can have long-term consequences for both public health and regulatory decision-making.

The growing availability of data and increasing reliance on statistical analysis in research have increased the need for researchers to have a strong understanding of statistical methods. However, many researchers have only basic statistical training and limited access to

statisticians [19,20]. As a result, they often encounter challenges in applying statistical methods correctly. In this study, we explored these challenges and misconceptions by examining the understanding of one of the most widely used statistical techniques in research: linear regression and its assumptions. While numerous tutorials in the literature focus on improving regression modelling [21], few studies directly evaluate authors' current practices. We aimed to understand the research-to-practice gap experienced by researchers and make recommendations to strengthen training and reporting guidelines.

## Study aims

This study explored the gap between statistical theory and practice by focusing on how statistical assumptions are reported and addressed in health-related research. Specifically, it evaluated the reporting practices for linear regression assumptions in 100 randomly selected papers published in *PLOS ONE* in 2019. Each paper was subjected to a post-publication review by statisticians to assess whether the authors reported checking assumptions, which assumptions were addressed, and how these were reported.

## Research questions

- What was the prevalence of publication author teams who reported in their manuscript that they had checked linear regression assumptions?
- Did author teams correctly check assumptions, and were misconceptions about linear regression assumptions evident in their reporting, as inferred from their methods and descriptions?
- What was the agreement of ratings for statistical assumptions made by different statisticians?

## Linear regression

This study was not designed to be an in-depth tutorial on linear regression but rather an overview so that the paper is accessible to non-statistical readers, for further reading on linear regression, see [22–24]. Linear regression models explain the relationship between a dependent variable and one or more independent variables by fitting a linear equation. In this equation, each independent variable is multiplied by a corresponding parameter, often referred to as the 'regression coefficient'. In its simplest form, when considering just one independent variable, average change in (Y) is predicted for each unit increase in (X), assuming that Y depends on X [25]. For example, how much does body weight change (on average) with a one-year increase in age? Linear regression allows the exploration of this relationship's direction and magnitude. While this paper focuses on linear regression assumptions, the second paper in this series examines reporting practices in linear regression, including key statistics to report and best practices for presentation, see Jones et al. [26].

Statistical inference is used to make conclusions about a population based on sample data. The main goal is to estimate parameters, such as regression coefficients, that describe relationships between variables. Since the entire population is typically not observed, these parameters are unknown and must be estimated using sample data. This paper presents sample-based regression equations with estimated values denoted by hats (e.g., $\hat{\beta}$ for regression coefficients). In these models, error terms represent the differences between observed and predicted values based on the population regression line. However, since error terms are unknown, we use residuals, sample-based differences to assess the model's assumptions.

Linear regression models are commonly fit using ordinary least squares (OLS) [22], which minimises the sum of the squared differences between observed values and their predicted values as represented by the regression line. This method identifies the "best-fitting line" by ensuring the total squared deviation between the actual data points and the predictions is as small as possible. Squaring these residuals serves two purposes: it prevents positive and negative deviations from cancelling out and provides a consistent measure of how far data points are from the line. Measuring variability using squared deviations is central to statistical concepts like variance, which is calculated as the average squared deviation between each data point and the mean. The square root of the variance is the standard deviation (SD), which describes the spread of data in the same units as the original values.

Linear regression, as described above, is, strictly speaking, a mathematical optimisation method where the best-fit line is estimated (Eq 1). However, we must quantify uncertainty around our estimates to draw statistical inferences, which requires certain assumptions. Assessing whether these assumptions hold is essential for reliable interpretation of results. One way to do this is by analysing the residuals, which are the differences between observed and model-predicted values (fitted values) (Fig 1). The adequacy of the model fit can be assessed by analysing the distribution of the residuals and identifying any patterns or systematic deviations from the regression model's assumptions.

$$\hat{Y}_i = \hat{\beta}_0 + \hat{\beta}_1 X_i + \hat{\epsilon}_i, \quad i = 1, \dots, N \quad (1)$$

In Eq 1, where $i = 1, \dots, N$ with $N$ being the total number of observations: $\hat{Y}_i$ is the dependent variable for the $ith$ observation. $\hat{\beta}_0$ is the $Y$-intercept, which is the estimated value of Y when X = 0. $\hat{\beta}_1$ is the slope, representing the average change in $Y$ for a one-unit increase in $X$. The error term is $\hat{\epsilon}_i$ for the $ith$ observation, representing the deviation of the observed value $Y_i$ from its predicted value based on the regression model. This equation can be extended to

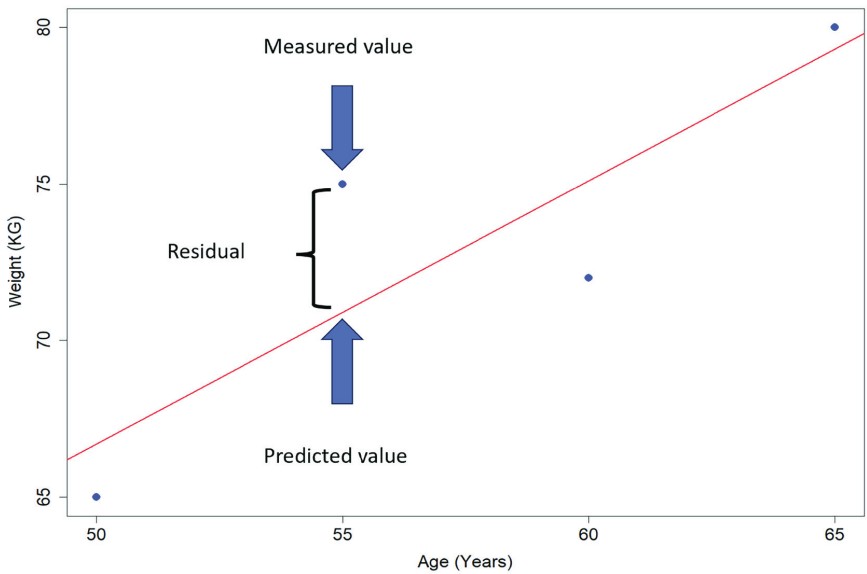

**Fig 1. Pictorial representation of a linear regression and residual, with the red line representing the line of best fit, the dots are the measured (observed) data.** The residual is the difference between the observed and predicted values.

incorporate multiple $X$ variables ($k$ parameters) as seen in Eq 2.

$$\hat{Y}_i = \hat{\beta}_0 + \hat{\beta}_1 X_{1i} + \hat{\beta}_2 X_{2i} + ... + \hat{\beta}_k X_{ki} + \hat{\epsilon}_i, \quad i = 1, ..., N \tag{2}$$

To undertake hypothesis tests and create confidence intervals that give realistic approximations of the underlying relationship between variables, it is assumed that residuals are: (i) normally distributed (with a mean of zero), (ii) the relationship between the outcome Y and the predictors is linear, (iii) have constant variance, and (iv) are independent. This can be visualised by plotting the residuals against the predicted values (Fig 2). If residuals are approximately normally distributed, standardising them allows problematic observations to be identified when values exceed $\pm 3$, as 99.7% of observations should fall within this range in a standard normal distribution. The terms predicted and fitted values may often be used interchangeably, but fitted refers to the values estimated by the model using the same data that was used to create the model. Whereas predicted is used in a broader sense, it may refer to the fitted values or data in a new dataset that was not used to create the model.

**Assumptions.**

1. Normality: The model residuals are assumed to be normally distributed (with a mean of zero).
2. Linearity: The mean of the dependent variable Y is assumed to change linearly as a function of the independent X variables and their associated parameters.
3. Homoscedasticity: The residuals are assumed to have constant variance across all values of X.
4. Independence: The residuals are assumed to be uncorrelated.

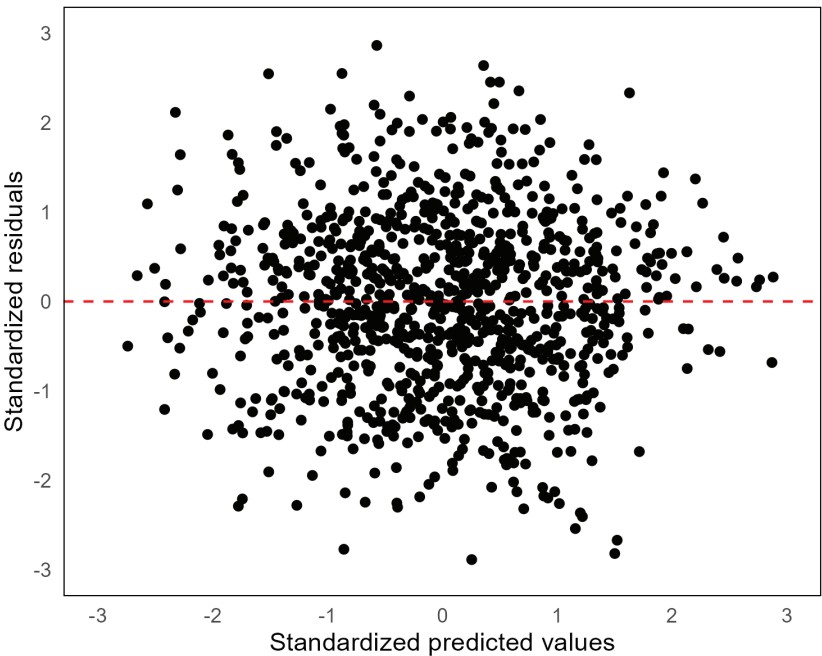

**Fig 2. Example of residuals from a linear regression model.** This example shows no clear violation of linearity, non-normality, or homoscedasticity, with the red line showing a hypothetical mean of zero, where there should be as many points above as below the line.

**Normality.**   Characteristics of a normal distribution include a symmetrical bell-curved shape around a mean, with mean, median, and mode all equal, and 95% of observations falling within approximately two (more precisely 1.96) standard deviations. Linear regression does not require the X or Y variables to be normally distributed, this assumption is only related to the residuals. Violation of normality does not necessarily lead to bias of regression coefficients, which depends on the sample size and degree of the violation. Non-normality of the residuals can lead to inaccurate estimates of p-values and confidence intervals and increased type I errors [27]. Regression models are generally not sensitive to small deviations from normality in the residuals, especially in large samples, but can be sensitive to heavy-tailed distributions or the presence of large outliers and influential points [28,29]. The best way to check this assumption is through examining the residuals with descriptive statistics, including examining if the mean and median of the distribution of the residuals are similar, exploring if skewness and kurtosis are within reasonable bounds, and visually through plots including histograms and quantile-quantile plots (Q-Q Plot) (Fig 3) [27]. The Q-Q plot is created by plotting quantiles from the data against quantiles generated from the normal distribution [30]. The points should follow an approximately diagonal line if the data are normally distributed.

**Linearity.**   The linearity assumption is that the relationship between the predictor terms of the model (which are typically the X variables or powers of them such as $X^2$) and the average change in the Y variable (dependent variable) are linearly related through model parameter/s, i.e. the regression coefficient/s. There is a common misunderstanding about linear regression models that each independent variable must relate linearly, which is taken to be a straight line, to the dependent variable, Y. This stems from the focus on simple linear regression, where there is only one X variable. In multivariable models, independent variables can be represented through multiple parameters, for example, age and age-squared, thus capturing more complex relationships, including splines and polynomials (e.g. quadratic, cubic, etc.). Even though $X^2$ is a nonlinear transformation of X (quadratic), the relationship remains

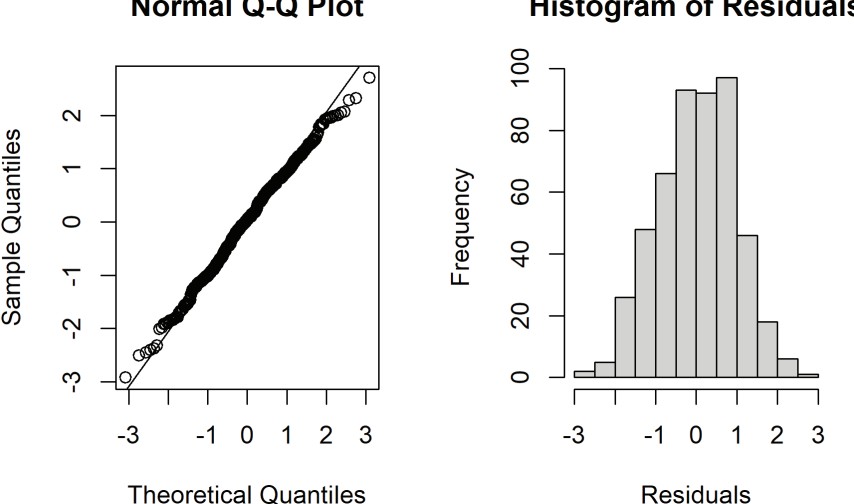

**Fig 3. Example of assessing the normality of the model residuals with plots, with points on the Q-Q plot generally following the diagonal line with a small deviation at the end and an approximate bell-shaped histogram, indicating that the residuals in this example are approximately normally distributed.**

linear in terms of the parameters. This is because, in linear regression, the expected value of Y is assumed to be a linear function of the parameters (coefficient/s) [31].

The linearity assumption can be visually checked through scatterplots of residuals against individual variables in the model as well as predicted values [27]. The residual plots should be examined for patterns, including curvature, which may indicate non-linear relationships. Fig 4 shows an example of linearity violation where a strong quadratic relationship is missing from the model. Understanding the underlying relationships in data is essential to resolving any problems. In the example above, the best solution is to identify the variable responsible for the issues and precisely model the polynomial relationship by properly adding a squared term to the model for that variable. Another solution may be to transform the data using the so called "ladder of transformations". For more detail, see [32].

**Homoscedasticity.** Homoscedasticity, also known as homogeneity of the variance, assumes that the residuals have constant variance and are distributed equally for all independent variables [33]. For example, in a linear regression model, where blood pressure depends on age. If the variance is constant or homoscedastic, then the variance of the errors will remain constant for all values of X. Therefore, the residuals will be equally spread around Y = 0, and the residual scatter should be similar at young and old ages. Suppose the variance is not constant (heteroscedastic). In that case, a plot of the residuals may show that being younger corresponds with a narrow range of low blood pressure, but as people age, blood pressure varies more widely. This may cause a funnelling shape in the residuals as seen in Fig 5 and may indicate that other variables explain blood pressure, such as several chronic conditions or smoking status.

Homoscedasticity violations can have serious consequences as they can bias the standard error, causing inaccurate significance values and confidence intervals, leading to increased type I error [34]. Diagnosis of heteroscedasticity is best made by visualising the residuals and predicted values using scatterplots. Still, it can also be assessed statistically with methods such as the Breusch Pagan test [35], although statistical tests are prone to over-detecting minor deviations in large samples and may lack power in small samples. As the cause of heteroscedasticity may not always be easily detected, it is essential to understand

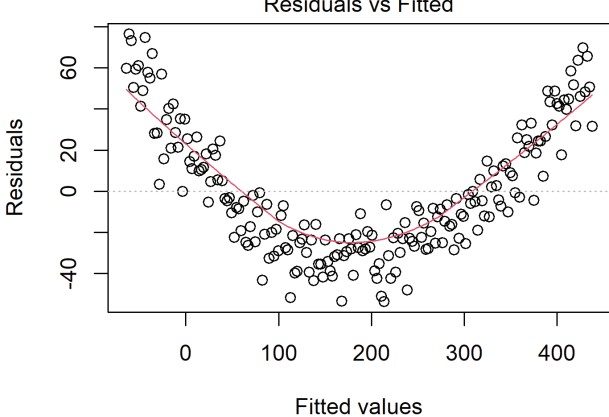

**Fig 4. Example of a non-linear relationship missing from the model that is detectable in the residuals.** The red line indicates a quadratic relationship between the residuals and fitted values.

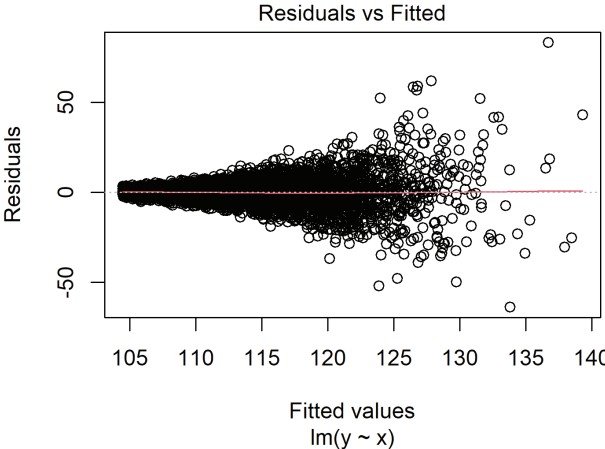

**Fig 5. Example of residuals displaying heteroscedasticity, where a funnel shape can be observed in the residuals instead of random scatter around zero (red line).**

the relationships in the data with both clinical understanding and plots. Remedies for heteroscedasticity may include weighted regression [36], robust standard errors [35], data transformation [22], including other variables in the model that improve prediction, or bootstrapping with a heteroscedasticity correction [34].

**Independence.** Linear regression assumes that each observation is independent of the other and that their residuals are uncorrelated. A commonly observed violation of independence generally involves repeated measures [27], for example, blood pressure measured over time in the same person. Measurements taken on the same participant (within-participant) are likely to be more similar than observations between participants. When generalising results to a population, treating correlated observations as independent can lead to an underestimation of the variance in the linear regression, making estimates appear more precise than they are in reality. This may lead to increased type I errors, making the accuracy of standard errors and confidence intervals questionable [37].

In addition to repeated measures, health research frequently involves other data structures where a correlation between observations (clustering) may be present [27] such as patients nested within doctors. The experience and ability of individual doctors may influence patient outcomes. Therefore, patients treated by the same doctor may not be independent. The independence of observations should be a part of the study design and sometimes may not require testing but should always be discussed. A lack of independence in the data can be visualised by plotting residuals by the individual (row number) to look for serial correlation (autocorrelation), when there are no violations; points should fall randomly around the zero line, which can be assessed using the Durbin–Watson test [27]. Suppose there is suspected clustering of the data. It can be examined by fitting an appropriate statistical method, such as a random effects model, to adjust for correlated observations. Therefore, the general remedy for independence violations is to use a method such as Linear Mixed Models (LMM) or General Estimating Equations (GEE) to account for the non-independence of data.

**Outliers and influential observations.** When undertaking statistical methods such as linear regression, it is important to identify influential observations that, if removed from the model, can substantially change the regression coefficients [33]. While the presence or

absence of outliers and influential observations is not an assumption of linear regression, they can potentially change results and may be the cause of assumption violations.

Outliers should not be routinely deleted. To minimise questionable research practices, the study protocol should address the management of outliers, such as data transformation, the use of robust regression, sensitivity analysis, bootstrapping, or variable truncation [38].

Two ways in which a single data point can affect the results of a model are when the observation is an outlier and/or has high leverage [39]. An outlier can be defined as where the response (Y) does not follow the general trend and falls outside the range of the other values. Outliers generally have large residual values with a sizable difference between the observed and predicted data. Leverage measures the distance between an observation's X value and the average X variable values in the data. Observations with extreme values of X are said to have high leverage [33]. Data points that display high leverage and/or are outliers have the potential to be influential but must be investigated to determine if they substantially change regression coefficients. A way to measure this change is known as Cook's Distance and is a combination of each observation's leverage and residual values [23].

In small datasets, large outliers can strongly influence regression coefficients, sometimes even changing the direction of the relationship. For example, consider a study measuring systolic blood pressure (SBP) and age in 10 adults. A good initial step in any linear regression is to visualise the data. In this case, SBP and age have a positive linear relationship, but a large outlier is observed for a 75-year-old with unexpectedly low SBP.

A sensitivity analysis (Fig 6) was performed to assess the impact of this observation. The estimated slope of SBP per unit increase in age nearly doubled, from 0.43 to 0.77, when the outlier was excluded. Visually, it is clear that including the outlier pulls the regression line downward at older ages, resulting in a poor fit to the overall trend. However, the outlier likely

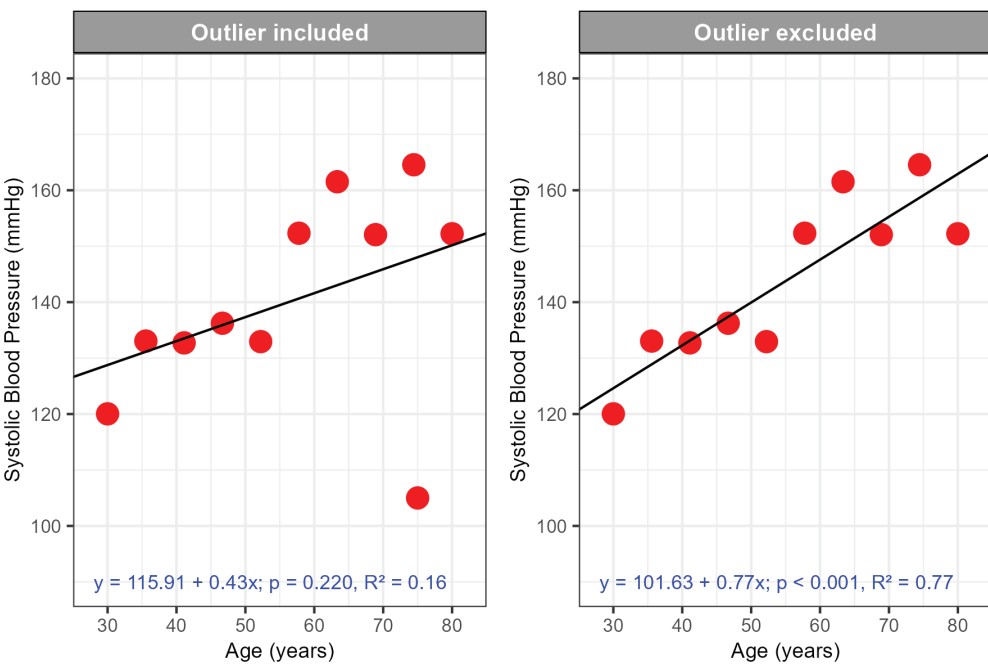

**Fig 6. Example of a sensitivity analysis showing linear regression results with outliers included and then excluded, using simulated data.**

represents a real observation, as some individuals naturally have low blood pressure throughout their lives. The large difference in $R^2$ and the regression coefficient between the two models suggests that they are capturing different patterns in the data. The model excluding the outlier has a higher-than-expected $R^2$, potentially overstating the strength of the relationship between age and SBP. This highlights the need for further investigation and adjustments, such as incorporating relevant clinical factors to improve model fit or applying data transformation or robust regression techniques.

Beyond assessing the impact of excluding the observation, we can examine diagnostic measures such as Cook's distance to determine whether this point has excessive influence on the regression model. One way to visualise this is through an influence plot, which maps standardized residuals against leverage, with Cook's distance contours overlaying the graph (Fig 7).

In this plot:

- Leverage (x-axis) measures how much an observation's X values differ from the rest of the data, meaning points further to the right have more potential to influence the regression line.
- Standardized residuals (y-axis) show how far an observation's Y's deviates from the model's prediction, helping identify extreme values.
- Cook's distance contours indicate the level of influence; points beyond the threshold values (often 0.5 or 1) suggest that these observations may disproportionately affect the model's estimates and should be examined further.

Observations that have both high leverage and large standardized residuals, particularly those exceeding the Cook's distance threshold, should be carefully examined. While some authors suggest a threshold of 4/n for small samples, others recommend a cutoff of >1 [22,23], particularly in larger datasets. However, whether >1 is a more or less conservative threshold depends on sample size, in small datasets, it may be too strict, potentially overlooking influential points, while in large datasets, it may be too lenient, allowing highly influential points to remain undetected. While not all high Cook's distance points are problematic, they signal cases where a single observation may significantly alter model estimates. These points should be investigated by checking data accuracy, verifying experimental conditions, and reassessing model assumptions to ensure the results remain meaningful.

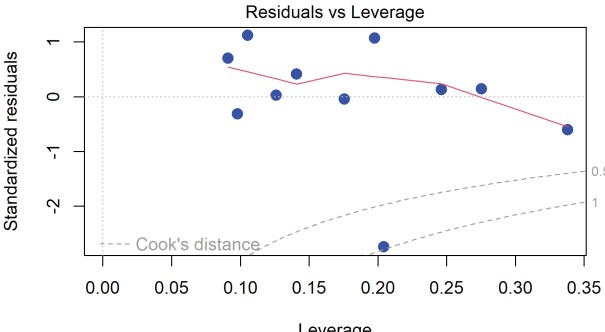

**Fig 7. Influence plot: showing the relationship between leverage, standardized Residuals, and Cook's distance, with thresholds at 0.5 and 1 indicating potentially influential observations.**

The appropriate response is context-dependent, there is no one-size-fits-all approach to handling potential outliers. If an unusually low value is caused by a known experimental error (e.g., a mistake in drug administration, such as only half the intended concentration being added), it would be reasonable to exclude the observation. However, if the low value represents natural variation, then data transformation or robust regression methods may be more appropriate to reduce its influence while preserving the integrity of the dataset. In small sample sizes, it is easy to assume that any data points deviating from the rest are outliers and should be removed. However, in smaller datasets, we might naturally observe less variation simply by chance, making unusual-looking points appear more extreme than they actually are. Removing these points risks eliminating natural variability causing biased estimates.

Regardless of the approach taken, transparent reporting is essential. Any decisions about outlier exclusion, transformation, or alternative modelling approaches should be clearly stated, along with the rationale behind them. Reporting results both with and without outliers (sensitivity analysis) provides a more complete picture of the data and ensures that findings are not driven by selective exclusion. By clearly documenting these steps, researchers can enhance the credibility, reproducibility, and interpretability of their findings.

**Multicollinearity.**   Multicollinearity occurs when two or more independent variables in a regression model are highly correlated, making it difficult to isolate their individual effects on the dependent variable [22]. This issue primarily affects the interpretability of the model, as the coefficients of correlated variables may become unstable, and their standard errors may be inflated. When multicollinearity is present, the shared variance among predictors reduces the amount of independent variance each variable uniquely explains in the model, making it harder to determine their true effects. Multicollinearity can be assessed using the variance inflation factor (VIF), which quantifies how much the variance of a coefficient is inflated due to correlation with other predictors [40]. Higher VIF values indicate that a variable shares variance with other predictors rather than contributing unique explanatory power to the model.

There has been debate about the threshold of VIF that indicates a problem, with values of 3, 5, and 10 commonly cited in the literature [40]. However, like all statistical cut-offs, VIF should be interpreted in the context of its effect on the model. For example, in a sensitivity analysis where a high-VIF variable is removed, multicollinearity may not be a significant concern if the standard errors and interpretation of variables remain stable across models and align with their univariate results. However, if the standard errors or direction of coefficients change drastically, even VIF values as low as three may indicate a problem.

An example where multicollinearity could be a concern is the relationship between age and years of experience when studying factors influencing doctors' adherence to medical guidelines. Older doctors generally have more experience, while younger doctors' experience is inherently limited by their age. Because of this strong correlation, distinguishing the independent effects of each variable may be challenging. If multicollinearity is problematic, age is often dropped in favour of years of experience, as the latter provides a more direct measure of expertise. However, the decision depends on the specific analysis and research question.

Another way to address multicollinearity without dropping variables is to use regularisation methods such as Lasso or Ridge regression. These methods help manage multicollinearity by penalising model complexity, shrinking coefficient estimates, and reducing sensitivity to collinear predictors. Lasso can also perform variable selection by setting some coefficients to zero, while Ridge retains all predictors but shrinks their impact [41].

## Materials and methods

The primary outcome is to understand authors' current reporting practices in published manuscripts regarding linear regression, with a focus on its assumptions. Previous studies show that the prevalence of reporting assumptions ranges from 0 to 13% [7,42], with assumptions most often being reported under 10% of the time. The prevalence of assumptions in this study was estimated by a random sample of papers meeting the search criteria of 'linear regression' from *PLOS ONE*.

### Sample size

A sample size of 100 papers was found to be adequate to detect a sample proportion of 0.05 (5%) using a two-sided 95% confidence interval with a margin of error of 5%. This sample size was calculated using a test for one proportion with exact Clopper–Pearson confidence intervals, using PASS [43]. For these papers, it was deemed feasible to recruit 40 statisticians (40 statisticians × 5 papers = 200 reviews), and from our experience and feedback during the development stage, having each statistician review five papers was manageable. Each paper was rated twice by two independent statisticians to increase the robustness of the results and provide data on the agreement in statisticians when checking assumptions.

### Question development

A set of questions was developed to understand current reporting practices for linear regression analysis. The questions were adapted from the Statistical Analyses and Methods in the Published Literature (SAMPL) regression guidelines [44]. A literature review was also used to identify common errors made by researchers when reporting linear regression, and a comprehensive list of 55 questions was developed to assess statistical quality. It was decided by the research team, consisting of three Australian accredited biostatisticians, to reduce the burden on reviewers by substantially reducing the number of questions. We focused on questions important to assessing assumptions and interpreting linear regression. The research team used an iterative approach to improve the wording of the questions by reviewing papers to understand issues reviewers may encounter. When these three statisticians were satisfied that the questions were appropriately worded, five independent experts (four biostatisticians and an epidemiologist) were given a briefing on the aims of this study and the questionnaire. They were asked to assess the questions and provide feedback on readability and length by examining two randomly selected papers. Their feedback was used to further reduce the questions to the current checklist of 30 items, which can be seen in S1 Table.

### Randomisation

The randomisation process of selected papers occurred in two steps, as described below.

**Paper selection and randomisation.** Papers which used the term 'linear regression' in the materials and methods section were selected from the 2019 issues of *PLOS ONE* using the "rplos" package in R [45]. Papers that matched the inclusion criteria (see below) were randomly ordered, and the first eligible 100 were selected. A complete list of Digital Object Identifiers (DOIs) of included and excluded papers is available for transparency [46]; a list of included papers can be seen in S2 Table.

**Inclusion criteria:**

- 'Linear regression' selected using the automated "searchplos" function within the "rplos" package, in the materials and methods section.

- *PLOS ONE* papers published between January 1st 2019 and December 31st 2019.
- Papers were selected which had health anywhere within the subject area, provided by "searchplos" function.
- With article type selected as Research Article, to exclude editorials etc.

**Exclusion criteria:**

- Linear regression models that have accounted for clustering or random effects e.g. mixed, multilevel models.
- Non-parametric linear regression, Bayesian, or other alternative methods to linear regression.
- Linear regression was not part of the primary analyses of the paper and was related to pre-processing the data or verifying an instrument or method of data collection e.g. a linear regression used to calibrate an instrument to a reference sample.

The exclusion criteria were used to make models comparable by excluding analyses that do not have the same assumptions or are more complex. The primary researcher read the papers, starting with the first in the random series until 100 papers met the inclusion but not the exclusion criteria. The number of papers excluded, and the reasons were reported. Due to the complexity of some papers, and to reduce the bias of excluding papers with poor quality, statisticians were allowed to exclude studies by answering that there were zero regressions in the paper despite the paper being selected for including linear regressions.

**Random allocation of papers to statisticians.** Allocating the papers to statisticians was achieved by using a one-way random design for the inter-rater reliability of the statistician. Fleiss [47] recommends that if there is no interest in comparing the mean of several raters, then a simple random sample of raters from the overall pool can be chosen. Hence, we randomly allocated papers using the following approach:

- Five papers were randomly allocated to each statistician.
- Papers were randomly reallocated to different statisticians, ensuring that no statistician was given the same paper twice.

**Statistician inclusion and recruitment.** We aimed to use qualified statisticians to review papers. Statisticians often come from diverse backgrounds, sometimes without formal statistics degrees, and researchers in ecology, psychology, and economics may identify as statisticians, data scientists and data analysts. This is also recognised by the professional bodies of statistics for accreditation [48]. Therefore, for inclusion in the study, statisticians were asked if they were employed or were previously employed as statisticians, data scientists or data analysts. Recruitment of statisticians occurred through targeted emails from information gained through organisational websites from within Australia and internationally, and more generically through Twitter, LinkedIn, professional societies such as the Statistical Society of Australia, universities, and other appropriate organisations.

Upon enrolling, statisticians were emailed a participant information sheet, the study protocol, the study questions, and an online link to five *PLOS ONE* papers to be reviewed, which can be accessed from [49]. Recruitment started in September 2020 and ended in June 2021, with the last participant completing the review in September 2021. Participants were sent automatic reminders every two weeks. The median time to completion was four weeks. Forty-six statisticians were recruited, and five withdrew due to changed circumstances or lack of time. One statistician had difficulty completing the online form, and was replaced.

**Ethics and consent.** This study was granted ethics approval from the Queensland University of Technology (QUT) Human Research Ethics Committee and was approved under the category Human, Negligible-Low Risk (approval number: 2000000458). Statisticians gave informed written consent by filling out and returning the participation form via email, which also asked if they wanted to be acknowledged in the publication. The *PLOS ONE* authors whose papers were studied were considered to have already consented as they agreed to a data sharing policy [50], which states that data may be used to validate and reproduce results.

## Measured outcomes

The outcome measures were the prevalence of statistical assumptions reported by authors in the randomly sampled papers, including normality, linearity, homoscedasticity, and independence, as well as reporting on outliers and multicollinearity assessment. Statistical raters first determined whether the authors reported checking assumptions and then selected from a predefined list of how assumptions were assessed and what was assessed. Since all responses were dichotomous, if an assumption was not ticked, it was assumed to be unreported. Statistical reviewers were prompted to check the supplementary by asking, "Was most of the detail for the assumption checks in the supporting information (appendix)?"

Statisticians also assessed whether authors reported checking for outliers and identified the action taken in a question, with response options such as *no action taken* or *sensitivity analysis*. Another question asked whether multicollinearity was assessed, with possible responses of *yes*, *no*, or *not required*. For further details on the questions asked, see S1 Table. The primary author, LJ, also gathered data about assumption checks, reviewed all papers and made qualitative notes about how assumptions were checked, misconceptions, and possible reasons for reviewer disagreements.

## Data analysis plan

This confirmatory study examined the reporting and quality of linear regression assumptions of published papers in the health and biomedical field. Reporting behaviours were described using frequencies and percentages, with Wilson 95% confidence intervals used to account for prevalence values close to zero.

The agreement of raters was described using observed agreement and Gwet's statistic [51], which performs well in situations of high agreement. Quadratic weighted Gwet's agreement was used for ordinal ratings. This weights disagreements according to the square of their distance on the scale and gives greater weight to larger disagreements compared to smaller ones. Assumptions of Gwet's agreement were considered and found to be acceptable, testing these assumptions is not required, as they are related to the design of the experiment, such as appropriate rating scales. Gwet's agreement was used for categorical data that was either nominal or ordinal. Gwet's agreement is less sensitive than Cohen's kappa to the distribution of ratings across categories (marginal distributions), which may be caused by statisticians rating different papers.

R version 4.4.2 [52] was used for all statistical analysis. This was a descriptive study with no formal hypothesis tests used, with prevalence and 95% confidence intervals reported, while statistical ratings were assessed for reliability. To increase transparency, the STROBE guideline was used for reporting cross-sectional studies [53].

## Calculating prevalence and reliability

This study was initially designed so the prevalence of individual assumptions could be estimated using the input of two statistical ratings, with the primary author (LJ) adjudicating disagreements. After the ratings for the first few papers were received, a test was carried out, where the primary author rated the papers and then checked agreement against the two raters; it was realised that due: (i) to the complexity and length of the papers and (ii) sometimes nuanced interpretation, a more comprehensive picture of prevalence was gained through all three ratings. Therefore, the prevalence of reporting behaviours was calculated using all three ratings. The primary author independently rated papers by filling in the survey and making notes on the PDF for the papers. The primary author adjudicated the difference between all three ratings by returning to the paper, making notes, and documenting each disagreement. Authors DV and AB were consulted when decisions were unclear. Reliability was calculated for the two statistical ratings, then the final prevalence rating was used as a gold standard to further assess the agreement of the two ratings separately. Missing data from reviewers was addressed in the reliability analysis by substituting the primary authors' rating.

## Results

In 2019, there were 1005 health research papers that reported using linear regression in the methods and materials section from *PLOS ONE*. Of these papers, 100 that met our inclusion criteria were randomly selected and sent out for statistical review (Fig 8). Reviewers could exclude papers by indicating there were no linear regression results reported in the paper. This was the case for ten papers; interestingly, there was little agreement among statisticians, with only one paper being excluded by all three statistical raters. After a review of these papers by the study authors, five papers were excluded due to having no reported regression results. Three of these papers reported the use of linear regression but did not report any individual results, two of which could be considered pre-processing; the other paper reported the use of ANOVA and linear regression but only reported the ANOVA results. The final two excluded papers used more complex methods, one using random effects and the other using Passing–Bablok regression. Therefore, 95 papers were considered in reviewing statistical reporting behaviours (Table 1), a majority of which were observational studies (84%) with human participants (77%).

Over half of the statisticians that agreed to review papers identified themselves as biostatisticians, with 83% of the sample having either a PhD or master's qualification and 53% having 10+ years of experience (Table 2).

## Reporting of linear regression assumptions and misconceptions

Of the 95 papers rated, 60 (63%) did not have any reporting of assumptions, 21 (22%) papers reported they checked one assumption, 9 (9%) reported on two assumptions, 5 (5%) checked three assumptions, with no author teams checking all four assumptions. Linearity was not required for 12 papers as they had no continuous independent variables; for these papers, only two checked one assumption (normality), with no other assumptions reported.

The questions initially asked if an assumption was checked, and then statisticians were asked to tick the boxes of how and what was checked. To avoid confusion with percentages, it was decided to keep the interpretation of how and why assumptions were checked at the level of papers rather than individual analyses within papers. Authors commonly reported checking continuous/quantitative data for normality but did not talk specifically about the

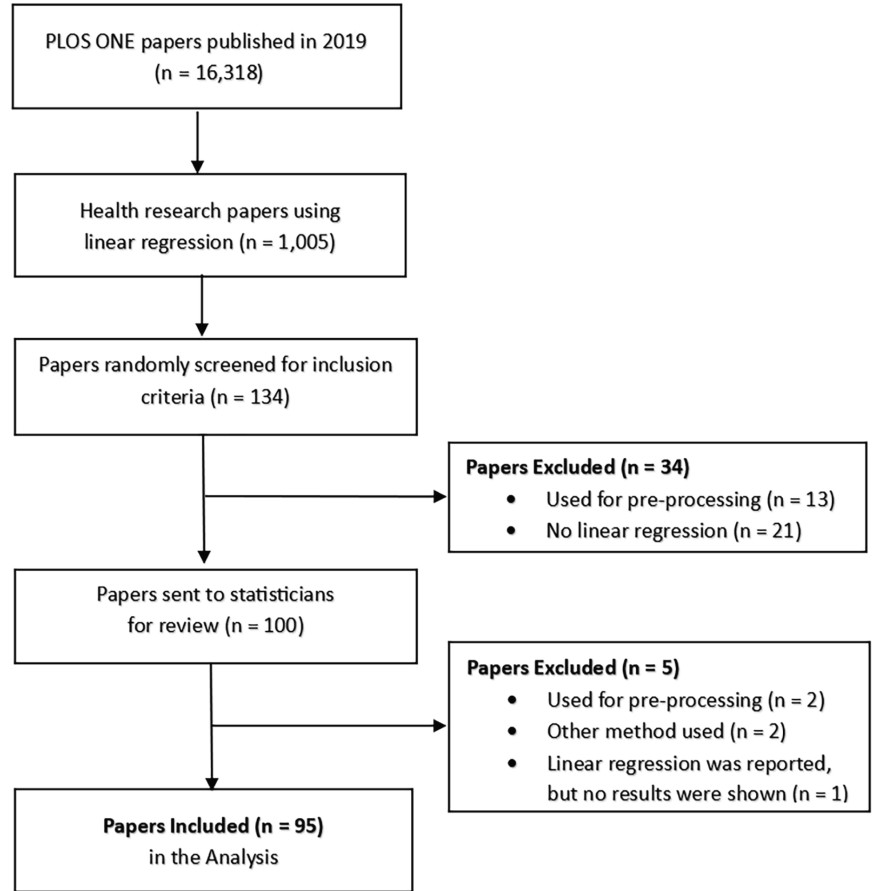

**Fig 8. Flow diagram of the included papers.**

regression analysis or residuals. Of the 28 (29%) papers that checked normality, five correctly checked the residuals (Table 3). Only three papers displayed some results of these checks, with partial reporting of results, with one paper reporting the optimal transformation result using a Box–Cox transformation and the other two showing box plots. A further six author teams presented box plots but did not mention normality. The same five papers that correctly checked residuals for normality were the only papers with a strategy for checking assumptions for linear regression.

Of the 83 papers that required linearity assessment, 15 (18%) directly assessed linearity, with a further 24 papers displaying scatterplots but not discussing them. Six authors who discussed linearity used scatterplots of the raw data to visualise relationships between variables; residuals were discussed in two papers; six papers used a test to assess linearity, with four authors fitting polynomials and two using splines. Homoscedasticity was discussed by 6 (6%) author teams, with most of these authors correctly checking this assumption using the residuals. Independence was addressed in 5 (5%) papers, while another 16 papers mentioned that their studies were cross-sectional but did not directly discuss independence.

Multicollinearity was assessed in 13 (14% CI: 8%, 22%) papers; 60 (63% CI: 53%, 72%) papers did not assess multicollinearity, which was not required for 22 (23% CI: 16%, 33%)

**Table 1. Description of included papers (N = 95).**

| Characteristic | n (%) |
|---|---|
| **Study Design** | |
| Observational | 80 (84%) |
| Experimental | 15 (16%) |
| **Participant Types** | |
| Human | 73 (77%) |
| Animal | 12 (13%) |
| Mix of animal and human | 3 (3.2%) |
| Mix of animal and plant | 2 (2.1%) |
| Other studies | 3 (3.2%) |
| Lab study with comparison to other studies | 1 (1.1%) |
| Environmental samples | 1 (1.1%) |

**Table 2. Descriptive statistics for statisticians (N = 40).**

| Characteristic | n (%) |
|---|---|
| **Role** | |
| Biostatistician | 21 (53%) |
| Statistician | 5 (13%) |
| Applied statistician | 7 (18%) |
| Data scientist | 2 (5.0%) |
| Data analyst | 2 (5.0%) |
| Other | 3 (7.5%) |
| **Highest statistical or mathematical education** | |
| PhD | 22 (55%) |
| Masters | 11 (28%) |
| Honours | 2 (5.0%) |
| Bachelor | 2 (5.0%) |
| Diploma | 1 (2.5%) |
| No formal education | 2 (5.0%) |
| **Years of experience** | |
| <5 years | 9 (50%) |
| 5–9 years | 0 (0%) |
| 10–19 years | 0 (0%) |
| 20+ years | 9 (50%) |
| Unknown | 22 |
| **How did you find out about this study?** | |
| Referred by a colleague | 18 (45%) |
| Professional society | 10 (25%) |
| Email | 10 (25%) |
| LinkedIn | 2 (5.0%) |

papers as they had only univariate models [26]. Correlation was used to assess multicollinearity in 6 papers, in four of these papers pairwise correlation was the only assessment; three of these mentioned moderate to high correlations without checking VIF. Another four authors mentioned multicollinearity checks but did not provide any details. Five authors reported they used VIF to check multicollinearity, with two authors using cut-off <5 another using <2.5; a further two reported in terms of the highest VIF to show there were no issues. No authors reported checking the standard errors or coefficient stability of regression models to determine if variables with high VIF or correlation were actually a problem.

**Table 3. Observed prevalence and 95% confidence intervals for statistical assumptions and outliers.**

| Variables | N | n (%) | 95% CI |
|---|---|---|---|
| Strategy for assessing linear regression assumptions? | 95 | 5 (5%) | 2%, 12% |
| Did the authors check the normality assumption? | 95 | 28 (29%) | 21%, 39% |
| What was checked with regards to normality? | | | |
| Unclear | 95 | 8 (8%) | 4%, 16% |
| Y variable | 95 | 18 (19%) | 12%, 28% |
| X variable | 95 | 8 (8%) | 4%, 16% |
| Sub groups of Y | 95 | 0 (0%) | 0%, 4% |
| Residuals | 95 | 5 (5%) | 2%, 12% |
| How was normality assessed? | | | |
| Unclear | 95 | 4 (4%) | 2%, 10% |
| Not described | 95 | 8 (8%) | 4%, 16% |
| Descriptive statistics | 95 | 4 (4%) | 2%, 10% |
| Plots | 95 | 10 (11%) | 6%, 18% |
| Statistical test | 95 | 9 (9%) | 5%, 17% |
| Did the authors check the linearity assumption? | 83 | 15 (18%) | 11%, 28% |
| How was linearity assessed? | | | |
| Unclear | 83 | 1 (1%) | 0%, 7% |
| Not described | 83 | 3 (4%) | 1%, 10% |
| Raw data | 83 | 6 (7%) | 3%, 15% |
| Plots | 83 | 9 (11%) | 6%, 19% |
| Residuals | 83 | 2 (2%) | 1%, 8% |
| Statistical test | 83 | 6 (7%) | 3%, 15% |
| Did the authors check the homoscedasticity assumption? | 95 | 6 (6%) | 3%, 13% |
| How was homoscedasticity assessed? | | | |
| Unclear | 95 | 0 (0%) | 0%, 4% |
| Not described | 95 | 2 (2%) | 1%, 7% |
| Raw data | 95 | 0 (0%) | 0%, 4% |
| Plots | 95 | 4 (4%) | 2%, 10% |
| Residuals | 95 | 4 (4%) | 2%, 10% |
| Statistical test | 95 | 0 (0%) | 0%, 4% |
| Did the authors check the independence of observations? | 95 | 5 (5%) | 2%, 12% |
| How was independence assessed? | | | |
| Unclear | 95 | 2 (2%) | 1%, 7% |
| Not described | 95 | 1 (1%) | 0%, 6% |
| Authors stated independent design | 95 | 2 (2%) | 1%, 7% |
| Raw data | 95 | 0 (0%) | 0%, 4% |
| Plots | 95 | 0 (0%) | 0%, 4% |
| Residuals | 95 | 0 (0%) | 0%, 4% |
| Statistical test | 95 | 1 (1%) | 0%, 6% |
| What did they do with respect to outliers? | 95 | | |
| Outliers not discussed | | 78 (82%) | 73%, 89% |
| Unclear | | 1 (1%) | 0%, 6% |
| No action taken | | 2 (2%) | 1%, 7% |
| Outliers removed from all analyses | | 10 (11%) | 6%, 18% |
| Sensitivity analysis | | 2 (2%) | 1%, 7% |
| Data transformation | | 0 (0%) | 0%, 4% |
| Bootstrapped | | 0 (0%) | 0%, 4% |
| Other | | 2 (2%) | 1%, 7% |

**Note:** N = Number of papers; n (%) = Prevalence; 95% CI = Wilson 95% confidence intervals.

Ten misconceptions regarding linear regression assumptions, outliers, and multicollinearity were identified through a combination of the prevalence of the questions and the main author's review of the papers (Table 4).

**Table 4. Common misconceptions for linear regression assumptions and outliers observed and inferred by this study.**

| Misconceptions | Recommendations |
|---|---|
| The normality assumption relates to the X and Y variables. | The normality assumption refers to the residuals rather than the X or Y variables. In a simple two-group example, if the means of the groups are different, the Y variable may not be normally distributed and possibly bimodal. A residual is a difference between what was observed and predicted by the model. There are expected to be some small, medium, and large residuals, but these residuals should be normally distributed. |
| Normality is the only important assumption. | Normality is the least important assumption; it becomes less critical with large sample sizes and is easily remedied by bootstrapping or data transformation. While residuals of a univariate model may not be normally distributed, adding other variables that improve prediction may remediate normality problems. |
| Normality needs to be checked with statistical tests. | Normality tests can either lack power in small samples or are too sensitive in large samples. In linear regression, residuals should be roughly normal and are best judged with a Q-Q plot rather than a statistical test. |
| Linear regression can only have variables with linear relationships | The linearity assumption does not necessarily mean that X itself is linearly related to Y. Instead, the relationship between the predictors (in which X variables can be represented through multiple parameters) and the dependent variable is linear in the parameters (coefficients). The most straightforward non-linear relationship is quadratic, with X and X-squared as independent variables. |
| Only the original data (X, Y) should be checked for linearity. | The original data should be plotted to understand linear and non-linear relationships, the residuals should also be plotted against predicted values to ensure no curvature patterns remain. |
| No need to check for equal variance (homoscedasticity) because there are no groups. | Linear regression models, t-tests and ANOVA (general linear models) all have the same assumption of equality of variance. While some researchers may realise checking variance (squared standard deviation) between groups is required, they may not be able to translate this to a regression context. Homoscedasticity can be examined by plotting the residual against the predicted values and looking for funnelling patterns. |
| Cross-sectional studies have independent observations. | The independence of observations is viewed by many researchers in the context of repeated measures, i.e., measurement of the same patient at two time points. There are frequently more complex study designs in health research, where patients may be clustered within hospitals or doctors. Study design should always be discussed, and when clusters occur, the correlation should be investigated using more complex methods such as linear mixed models. |
| All outliers should be removed from the model. | Outliers should only be removed if they are data errors, e.g., implausible values. Removing outliers artificially reduces the variance and may exaggerate results. The presence and effect of outliers should be investigated and discussed. A generally useful solution is a sensitivity analysis allowing the impact on the model to be assessed, other remedies may include bootstrapping or data transformation. |
| Only pairwise correlations should be used to assess multicollinearity. | While pairwise correlations can indicate collinearity between two variables, they do not account for cases where a variable shares variance with multiple predictors simultaneously. Even if pairwise correlations are low/moderate, a variable may still be collinear with a combination of other variables. Therefore, variance inflation factor (VIF), changes in standard errors (SE), and regression coefficients should always be examined when assessing multicollinearity in regression models |
| Variables with high correlation or VIF should be immediately dropped. | While high correlations and VIF may indicate multicollinearity issues, relying solely on rules of thumb to define problematic variables is not sufficient. It is important to assess whether multicollinearity is actually affecting the model by examining changes in regression coefficients and standard errors. If an issue is identified, appropriate methods such as regularisation or variable selection based on clinical knowledge should be considered. |

## Agreement of statistical raters

The agreement between statistical raters on assumptions and outliers was high (Table 5, for full reporting, see S3 Table), with observed agreement of over 80% for all assumptions except independence, which had a slightly lower agreement of 78%. When considering agreement by chance for independence, the Gwet's statistic was 0.69; while this is still regarded as

**Table 5. Agreement and reliability of statistical raters.**

| Variable | Rating 1 vs Rating 2 | | | Rating 1 vs Prevalence | | | Rating 2 vs Prevalence | | |
|---|---|---|---|---|---|---|---|---|---|
| | Agree | Gwet | 95% CI | Agree | Gwet | 95% CI | Agree | Gwet | 95% CI |
| Normality | 88% | 0.81 | 0.70, 0.93 | 91% | 0.84 | 0.74, 0.95 | 92% | 0.86 | 0.76, 0.96 |
| Linearity | 89% | 0.85 | 0.75, 0.96 | 92% | 0.88 | 0.79, 0.97 | 90% | 0.87 | 0.77, 0.96 |
| Homoscedasticity | 100% | | | 98% | 0.98 | 0.94, 1.00 | 98% | 0.98 | 0.94, 1.00 |
| Independence | 78% | 0.69 | 0.55, 0.83 | 91% | 0.89 | 0.81, 0.96 | 83% | 0.78 | 0.66, 0.90 |
| Outliers | 83% | 0.82 | 0.74, 0.91 | 91% | 0.90 | 0.84, 0.96 | 86% | 0.86 | 0.78, 0.93 |

**Note:** Agree = Observed agreement, Gwet = Gwet agreement coefficient; 95% CI = Gwet 95% confidence intervals.

good agreement [54], it is an arbitrary threshold and was lower than expected given expert raters. Reviewing the disagreements, a number of authors stated that their studies were cross-sectional without discussing independence and potential clusters within the data. Therefore, simply mentioning a cross-sectional study design was not considered an assumption check. The other main reason for disagreement among raters was that some papers contained plots without any discussion of assumptions, these were counted but not considered an assumption check, as it is possible to show a scatterplot with a non-linear relationship present. When comparing statistical ratings to the final calculated prevalence (gold standard), these were generally higher than between raters, indicating good overall reliability in the study. While there was no missing data in the assumptions and outliers section of the questionnaire, four of the 190 reviews were rated by one of the statisticians to have no linear regression, so they were replaced by the primary authors' rating for the reliability analysis.

Poor agreement between raters was observed for multicollinearity, with a raw agreement of 65% and Gwet 0.43 CI: 0.23, 0.62; the low agreement was likely due to poor reporting [26]. It was often difficult to determine whether papers should be placed in no or not required categories, as it was often unclear whether reported results were univariate or multivariable models [26].

Statisticians were asked to rate the statistical quality of each paper on a Likert scale with one representing very poor and five indicating very good. Gwet's using quadratic weights showed good agreement (0.74 CI:0.64, 0.83) between raters. After averaging the ratings, the mean ratings for papers were 2.5 SD: 0.8, indicating that statisticians rated the statistical quality between poor and fair.

## Discussion

Results showed that only 37% of authors checked any linear regression assumptions; this was similar to a review of papers by Real et al. [55], who examined the quality of reporting for multivariable regression models in observational studies and found of the 77 papers using linear regression, 39% reported testing assumptions. However, the authors did not provide details on which assumptions were tested. In our study, 29% of author teams suggested they checked for normality, only 5 of these papers mentioned residuals, and 19% wrongly checked the Y variable. This common statistical misconception about normality was higher in our study than in a previous study by Ernst and Albers [25], who assessed 172 papers in clinical psychology using linear regression and found that 4% mistakenly assessed the original variables' normality rather than the models' residuals. The higher prevalence observed for this misconception in our study may be due to lower statistical literacy in general health and biomedical areas, with health professionals often having completed one introductory statistics course [5]. In contrast, most psychology degrees have higher levels of statistical training. However, Ernst and Albers [25] indicate that reporting practices for regression assumptions

in clinical psychology journals were generally poor, with only 2% of papers being transparent and correct.

A study with a more comparable population by Fernandez-Nino and Hernandaz–Montes [7] assessed 108 papers in the health research journal *Biomedica* between 2000 to 2017. The authors used a detailed checklist reviewing statistical modelling, including statistical assumptions. This study concluded that 22% of papers mentioned any statistical assumptions, with 13% reporting checking normality, 3% linearity, 8% homoscedasticity, and 8% independence, with only 9% having a strategy to explore assumptions. Another study reviewing ANOVA reporting practices in three psychology journals in 2012 [42], found that 94% of papers did not provide statistical information on assumption tests. Only 5% of authors assessed normality, and 3% homogeneity of variance, with none discussing independence. A study in the Orthopaedic literature [56] found that no papers checked all linear regression assumptions with 25% (29/79) checking at least one assumption. We observed similar results as other studies with low reporting of independence (5%) and homoscedasticity (6%) but had a higher prevalence of discussing normality (29%) and linearity (18%). While this higher prevalence may simply be sample-to-sample variance, it may suggest that authors are starting to get the message that assumptions need to be checked, as journals increasingly use reporting guidelines. Like other studies, only a few author teams correctly checked assumptions or provided any details of assumption checks.

Questionable research practices occur when outliers are selectively removed, which may produce a statistically significant result that would otherwise not be significant [57]. It has been found in much of the health literature that identifying influential observations is either entirely missing or poorly assessed [38,58]. Our results confirmed that reporting outliers needs improvement, with no discussion of outliers in 78 (82%) papers, with 10 papers removing outliers from all further analyses with only two papers using a sensitivity analysis. This was higher than Fernandez–Nino and Hernandaz–Montes [7] who reported 4% of 113 papers reviewed in *Biomedica* mentioned outliers, but similar to results reported by Valentine et al. [59]. In response to the reproducibility crisis in psychology, Valentine et al. [59] conducted a study examining the reporting of outliers at two-time points, firstly in 2012 at the beginning of the crisis (poor practice occurred before this period, but the extent of the problem was formally explored in 2012), and in 2017. A total of 2235 experiments were analysed, with authors concluding there had been an increase in reporting of outliers in psychology from 16% to 25%, but reporting practices remained poor.

Our study found similarly low reporting of multicollinearity, with only 13 out of 73 papers assessing it when it should have been considered. This aligns with Fernandez–Niño and Hernández–Montes [7], who found that 15% (17/113) of models in *Biomedica* assessed collinearity. Norstrom et al. [60] reported an even lower rate, reviewing 41 public health studies and finding that only one tested for collinearity. This is more consistent with Vatcheva et al. [61], who searched the epidemiological literature in *PubMed* from 2004 to 2013 and found that only 1 in 100 regression papers mentioned collinearity or multicollinearity. Our results indicated that the authors often check correlation, or VIF, without a practical understanding of the effects of multicollinearity, with the authors falling back on the rules of thumb without actually assessing the effect of the correlation. Another author mistakenly used stepwise modelling procedures, believing this approach would address multicollinearity by selecting the most significant variables. However, they did not recognise that multicollinearity causes regression coefficients and standard errors to become unstable, leading to potentially unreliable results. In such cases, authors would benefit from refining the model by drawing a causal diagram to identify which variables need to be adjusted for. Ideally, this should be done before the analysis stage.

## Peer and statistical review

Although peer review is considered the most trustworthy way of selecting manuscripts for publication and improving the quality of papers in medical journals, Cobo et al. [62] advise that there is very little evidence to support this view. Altman [63] suggested that reviewers are often no more knowledgeable than the authors and recommended that statistical reviewers be used to reduce errors and improve quality. In the only randomised controlled trial in assessing the effectiveness of statistical review [62], papers were allocated into four groups (1) clinical reviewers (control group); (2) clinical reviewers plus a statistical reviewer; (3) clinical reviewers with a checklist; and (4) clinical reviewers plus a statistical reviewer and checklist. This study concluded a measurable improvement in the quality of papers with statistical reviewers, but no improvement in quality was observed in the checklist group. Statistical review results in important changes to manuscripts above and beyond average review about 60% of the time and is essential in improving the quality of published manuscripts [64]. The generally low ratings for papers by statisticians in our study indicate that authors would have benefited from statistical reviews pre-publication and can still benefit from feedback from this post-publication statistical review. We found that the methods sections were often unclear and did not have a detailed account of assumptions checked. While it is encouraging that many researchers are using scatterplots to visualise data, the discussion of assumptions remains sparse.

It is recommended that statistical reviewers should always be part of editorial teams. The method (linear regression) reviewed here is commonly used in the health field, and assumptions are relatively straightforward to interpret. If we extrapolate, the problems are expected to be greater for more complicated methods such as mixed models, structural equation models, etc. It is recognised that the volume of papers going through journals means that a statistician will only view a small proportion of papers going through to publication. Therefore, journals should invest in basic statistical resources for researchers and reviewers. While *PLOS ONE* already recommends the use of the *SAMPL* guidelines [44], which include confirming that regression assumptions are met, our research found low adherence of following these guidelines for either reporting assumptions or general reporting [26]. This suggests that reviewers could encourage compliance by providing direct links to the guidelines when poor reporting is identified. A limitation of the SAMPL guidelines is that they are text-based; incorporating visuals and tables could enhance clarity and highlight correct reporting practices.

There is also an opportunity to implement automated tools to search for tests and match appropriate assumptions in documents [65,66]. While this approach should not replace human reviewers, it can complement them, save reviewers time, and produce automated feedback to researchers, directing them to statistical resources.

Researchers discussing assumptions would be a big improvement on current practice. Detailed assumption checks can be placed in supplementary tables and plots. Journal editorial policies should also be considered. In most journals, page/word limits result in relatively limited space for statistical methods, although some of this can go into supplemental materials. It is possible that some teams did the appropriate checks but chose to avoid reporting on them due to reducing complexity (saving space) or the perception that doing so could make the review process more difficult. *PLOS ONE* does not have page or space limits, so this may be less of a consideration in this case. However, the author's normal reporting practice would be expected to affect how statistical sections are reported. It is recommended that journals focus on good scientific practices for statistical sections, which focus on describing what was done in enough detail so that another researcher could reproduce the results.

## Strategies for teaching statistics

In teaching statistics to health professionals, it can be tricky to get the balance right between too much theory and too little. Introductory statistical courses may compound problems for health professionals, which are often taught in a cookbook manner, where there is no emphasis placed on investigating the appropriateness of statistical methods [67,68]. Our results reinforced this view with many papers checking only normality, often with generic statements about continuous variables. Several authors used combinations of univariate non-parametric tests followed by linear regression to do multivariable modelling without commenting on assumptions. These results suggest that importance needs to be placed on underlying statistical theory rather than teaching statistics as an isolated series of tests so that methods can be put in context and better understood by relating them to other methods.

First-year statistical courses often emphasise t-tests, ANOVA and regression individually with well-behaved data. Students are then offered the alternative of the non-parametric test if data is not normally distributed. This basic understanding of assumptions of parametric tests is pervasive. Many researchers are unaware that t-tests, ANOVA and linear regressions can be seen in a general linear model framework [69], where X variables can be either categorical or continuous. This knowledge is vital in selecting the correct choice of statistical tests, and assumptions are related to the residuals rather than the raw X or Y variables. Researchers often feel comfortable testing for normality and using non-parametric tests because it gives a binary answer, and there is comfort in following exact rules. While there is nothing wrong with using non-parametric tests, they often lack power, and the choice of descriptive statistics should fit in with the overall goal of the analysis. If the purpose is multivariable modelling, using non-parametric statistics for the univariate step does not make sense. It is recommended that health professionals be taught more holistically with a bigger picture of 'everything is a regression' [70], emphasising statistical thinking where students become more comfortable with uncertainty, and statistical assumptions be taught in the broader overview of modelling rather than a narrow univariate sense.

## Limitations

*PLOS ONE* is a mega-journal crossing many disciplines but may not represent all journals. Therefore, this study may not be generalisable to all fields. Papers were randomly selected using the term linear regression; this may be biased toward authors with enough knowledge to identify the correct name. Although the bias is unknown, naming conventions may also be field-specific and unrelated to quality. Finding these papers would require the researchers to read a wide selection of papers that would be time-consuming and may not yield many additional papers. In scoping this project, an automated search of *PLOS ONE* was created to identify the term 'regression'. Then papers identifying other forms of regression (e.g., logistic or Poisson regression) were excluded. Although this was effective, it excluded papers using linear regression with other methods. As it is common for authors to undertake multiple forms of analysis in papers, it was decided that a simple approach of searching for linear regression would be more representative of papers in general.

A limitation of this study is its scope, as it focuses on a specific set of papers within a single journal focused on health research. Expanding the analysis to include papers from multiple journals or diverse research fields would provide a more comprehensive understanding of statistical reporting practices and assumption checking across disciplines. Future research should consider incorporating a broader range of studies to enhance generalisability and allow for cross-disciplinary comparisons.

Including 40 statistical raters potentially reduced rater bias but may have increased variability in some questions. Using two trained statistical raters may have reduced this variability. Still, the authors believe the design used was more reflective of real-world statistical reviews of papers and is, therefore, generalisable. This bias was explored by calculating agreement between the final prevalence score and each set of ratings, which tended to be higher than between the two sets of ratings, indicating while there was some variability, the individual statistical ratings were reflective of the overall results.

## Future research

Future research will investigate whether statistical assumptions influenced reported results by reproducing analyses, regardless of whether authors explicitly stated they checked assumptions. This will involve systematically re-analysing data to determine whether violations of assumptions, such as heteroscedasticity, multicollinearity, or non-normality of residuals, lead to different statistical conclusions. We aim to assess the extent to which unchecked or improperly addressed assumptions may have affected the validity of reported results. Additionally, we will examine whether reviewers provided statistical recommendations during peer review and assess whether these suggestions contributed to improvements in the manuscript.

## Conclusions

This study contributes to the growing field of meta-research by critically examining current statistical practices and highlighting ten common misconceptions researchers make about linear regression assumptions, outliers and multicollinearity. These misconceptions reflect a disconnect between theoretical statistical knowledge and its practical application in research. Addressing this gap is not merely an academic exercise but a necessity for ensuring the accuracy and reliability of research findings, particularly in fields like health and medicine, as flawed analyses can be used as evidence to support ineffective or harmful care, ultimately impacting patient outcomes.

To bridge this research-to-practice gap, it is essential to rethink how statistics is taught. Instead of presenting statistical methods as a collection of disconnected tests, teaching should adopt a holistic approach that frames most statistical analyses within the regression paradigm. This perspective fosters a deeper understanding of statistical concepts and equips researchers with the tools to apply these methods correctly across diverse scenarios. By promoting statistical literacy, researchers are more likely to produce accurate analyses, reducing the risk of misinterpretation and subsequent harmful applications in clinical settings.

Reviewers, as gatekeepers of scientific rigour, also play a pivotal role in ensuring the quality of published research. However, many reviewers lack sufficient statistical training to assess methods critically. Giving reviewers basic statistical education, access to resources, and automated tools may help them identify common errors, flag omissions, and offer constructive feedback. For example, a reviewer who understands the importance of verifying linear regression assumptions is better equipped to identify flawed analyses that might otherwise go unnoticed.

Journal editorial practices should similarly prioritise transparency and reproducibility. Current editorial policies often emphasise brevity, leading to the omission of critical details about statistical methods. This not only hampers reproducibility but also obscures potential flaws in the analysis that could affect real-world applications. Journals should encourage comprehensive reporting of statistical analyses, including the steps taken to check and address assumptions. This is vital for medical research, in particular, as decisions based on

poorly reported or misinterpreted studies can lead to misguided clinical guidelines, ineffective interventions, and harm to patients.

Ensuring rigorous statistical practices has a direct link to patient outcomes. Inaccurate or poorly analysed data can cascade to healthcare decision-making, resulting in wasted resources, ineffective treatments, and reduced trust in scientific research. By improving how statistics is taught, how research is reviewed, and how journals emphasise reporting, the scientific community can improve the reliability of research findings and their application to clinical practice.

## Supporting information

**S1 Table Formatted linear regression statistical questionnaire.**
(PDF)

**S2 Table DOI and titles for included papers.**
(CSV)

**S3 Table Full reporting of agreement and reliability for statistical raters.**
(DOCX)

## Acknowledgements

We acknowledge all the statisticians (named and not named) who kindly gave up their time to contribute to this publication by reviewing papers, including: Ingrid Aulike, Peter Baker, Brigid Betz-Stablein, Enrique Bustamante, Taya Collyer, Susanna Cramb, Alanah Cronin, Laura Delaney, Zoe Dettrick, Eralda Gjika Dhamo, Des FitzGerald, Peter Geelan-Small, Edward Gosden, Alison Griffin, Jenine Harris, Cameron Hurst, Kyle James, Helen Johnson, Jessica Kasza, Karen Lamb, Stacey Llewellyn, James Martin, Miranda Mortlock, Satomi Okano, Alan Rigby, Michael Steele, Megan Steele, Jacqueline Thompson, Simon Turner, Michael Waller, Kevin Wang, Jace Warren, Natasha Weaver, Lachlan Webb, and Janet Williams.

## Author contributions

**Conceptualization:** Lee Jones, Adrian Barnett, Dimitrios Vagenas.

**Data curation:** Lee Jones.

**Formal analysis:** Lee Jones.

**Funding acquisition:** Lee Jones.

**Investigation:** Lee Jones.

**Methodology:** Lee Jones, Adrian Barnett, Dimitrios Vagenas.

**Project administration:** Lee Jones, Dimitrios Vagenas.

**Resources:** Lee Jones, Dimitrios Vagenas.

**Software:** Lee Jones.

**Supervision:** Adrian Barnett, Dimitrios Vagenas.

**Validation:** Adrian Barnett, Dimitrios Vagenas.

**Visualization:** Lee Jones.

**Writing – original draft:** Lee Jones.

**Writing – review & editing:** Lee Jones, Adrian Barnett, Dimitrios Vagenas.

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
