## [Decision Letter · Decision Letter 0]

PONE-D-24-05674Common misconceptions held by health researchers when interpreting linear regression assumptions, a cross-sectional studyPLOS ONE

Dear Dr. Jones,

Thank you for submitting your manuscript to PLOS ONE. After careful consideration, we feel that it has merit but does not fully meet PLOS ONE’s publication criteria as it currently stands. Therefore, we invite you to submit a revised version of the manuscript that addresses the points raised during the review process.

We look forward to receiving your revised manuscript.

Kind regards,

Mohamed R. Abonazel, Ph.D.

Academic Editor

PLOS ONE

***Comments from PLOS Editorial Office:** One or more of the reviewers has recommended that you cite specific previously published works. Members of the editorial team have determined that the works referenced are not directly related to the submitted manuscript. As such, please note that it is not necessary or expected to cite the works requested by the reviewer. *

Journal requirements: When submitting your revision, we need you to address these additional requirements. 1. Please ensure that your manuscript meets PLOS ONE's style requirements, including those for file naming. The PLOS ONE style templates can be found at https://journals.plos.org/plosone/s/file?id=wjVg/PLOSOne_formatting_sample_main_body.pdf and https://journals.plos.org/plosone/s/file?id=ba62/PLOSOne_formatting_sample_title_authors_affiliations.pdf. 2. For ease of reference, please provide a list of the papers assessed in this study. This can be as a file uploaded as supporting information, or as a link included in your revised manuscript text to the location in a repository where this file can be accessed. 3. Thank you for stating the following financial disclosure:  [There was no cost associated with this research except for attending conferences. These costs were covered by the primary author's PhD allocation from the health faculty, Queensland University of Technology, and scholarships. The Statistical Society of Australia (SSA) and the Association for Interdisciplinary Meta-research & Open Science (AIMOS) supported the primary author with travel grants to attend their respective conferences. These scholarships did not influence the results of the study.].  Please state what role the funders took in the study.  If the funders had no role, please state: ""The funders had no role in study design, data collection and analysis, decision to publish, or preparation of the manuscript."" If this statement is not correct you must amend it as needed. Please include this amended Role of Funder statement in your cover letter; we will change the online submission form on your behalf. 4. We are unable to open your figure file [Fig 4.eps and Fig 6.eps]. Please kindly revise as necessary and re-upload.

Reviewers' comments:

Reviewer's Responses to Questions

**Comments to the Author**

1. Is the manuscript technically sound, and do the data support the conclusions?

Reviewer #1: Yes

Reviewer #2: Yes

Reviewer #3: Yes

Reviewer #4: Yes

Reviewer #5: Yes

Reviewer #6: Yes

Reviewer #7: No

Reviewer #8: Yes

Reviewer #9: Yes

2. Has the statistical analysis been performed appropriately and rigorously? 

Reviewer #1: Yes

Reviewer #2: Yes

Reviewer #3: Yes

Reviewer #4: Yes

Reviewer #5: Yes

Reviewer #6: Yes

Reviewer #7: No

Reviewer #8: Yes

Reviewer #9: Yes

3. Have the authors made all data underlying the findings in their manuscript fully available?

Reviewer #1: Yes

Reviewer #2: Yes

Reviewer #3: No

Reviewer #4: Yes

Reviewer #5: Yes

Reviewer #6: Yes

Reviewer #7: Yes

Reviewer #8: Yes

Reviewer #9: Yes

4. Is the manuscript presented in an intelligible fashion and written in standard English?

Reviewer #1: Yes

Reviewer #2: No

Reviewer #3: No

Reviewer #4: Yes

Reviewer #5: Yes

Reviewer #6: Yes

Reviewer #7: No

Reviewer #8: Yes

Reviewer #9: Yes

5. Review Comments to the Author

Reviewer #1: The manuscript is based on very unique idea and it is the demand of time to perform study like this. the article is written in very intelligent fashion and all data is available for results reproduction.

Reviewer #2: I recommend accepting this paper, but after making the following modifications:

1- I think that the abstract needs improvement.

2- I think that some recent papers related to this research should be mentioned.

3- At the conclusion of this work, the limitations of this research should be mentioned.

Reviewer #3: Dear Editor, thank you much for inviting me to review this captivating and very important scientific work. I also appreciate the authors this work.

Introduction:

This section is looking general and lacks the study’s aims, the linear assumptions. The authors would elaborate the assumptions of linear regression analysis. So the authors should revise and rewrite this section.

Research questions: Tense error.

Materials and Methods: The first paragraph would be included under introduction section. What it implies in method section?

This section does not show the study design in detail, the study variables and measurement. And also it does not data control measures undertaken.

Results: The results would be reported according the study’s questions (under three sub-headings)

Reviewer #4: In my opinion, the paper offers a good review study. I suggest the following modifications:

1. In both equations (Eqs. 1 and 2), the use of symbols with “hat” is incorrect. Why do you use "hat" in the regression equation before estimation? Please check any book on regression models and correct this.

2. The statistical assumptions of the model need to be corrected.

3. One of the important problems in multiple regression is multicollinearity. Why didn't you mention it in this research? Authors should check the multicollinearity test in each paper from the study sample.

4. Add a list of the titles and DOI numbers of papers used in this study to a supplementary file.

Reviewer #5: General comments:

This manuscript contains the results of a reporting review of linear regression usage in PLOS ONE papers published in 2019.

This paper follows in a long line of similar papers that evaluate the usage or reporting of statistical methods. My opinion is that these reviews should be performed regularly and across different research areas. I found this review to be of a high technical standard and feel that it meets the PLOS ONE publication criteria. I have a few general comments below as well as a couple specific comments.

A main point is that information about assumptions being met is predicated on there being the space for and, in my opinion, the desire or requirement of this information to be reported. Regarding the former, in theory, PLOS ONE is a good journal to study because there is no word limit and including supplementary materials is allowed. But, because PLOS ONE is generally not the initial submission choice for authors, some manuscripts may have been initially constructed for journals with word and figure/table limits. I bring this up because, in articles written to a 3,000 or 3,500 word limit, there is rarely the space for a thorough evaluation of the assumptions. In addition, as other methods have become more detailed (e.g., study designs, laboratory methods) the amount of space devoted statistical methods have gotten squeezed out.

This is generally why reviews such as this are performed on more prestigious journals because authors focus on those journals and, hence, the content is more uniform across articles. In addition, more prestigious journals more often have standard statistical review practices. PLOS ONE does have a group of statistical reviewers (of which I am one!), but the sheer volume of papers means we cannot review all of them.

As you mentioned in your limitations, the results are not generalizable to all fields. I will also mention that different fields vary substantially in how and what statistical methods are taught. This is another reason why reviews in field specific journals can be more useful because the typical curriculum for students in that field can be critiqued and recommendations can be made. That can't be done here with the wide array of articles in PLOS ONE.

In addition, reviewers may vary in their statistical knowledge. Although linear models are the cornerstone of statistical modeling, experience may vary substantially. For instance, I have been an applied statistician for over 20 years, but rarely use linear models.

Ultimately, I think reporting is only going to change if journals require it. CONSORT, STROBE, and other reporting guidelines have shown that better reporting can happen. But, authors are not going to change unless failing to report how model assumptions were evaluation will preclude publication. Thus, I think the target audience for your review should be journal editors. Maybe for your next endeavor you can take this article and draft guidelines for the reporting of regression models.

At any rate, I think this is a quality review and I recommend it for publication.

Specific comments:

1. (line 91) The rplos package has been removed from CRAN and archived on GitHub. I think this should be noted in your manuscript since this will make it harder for this research to be reproducible.

2. (line 457) Why is "Questionable Research Practices" capitalized?

3. It wasn't clear to me whether the statisticians read any supplementary materials to evaluate the methods.

Reviewer #6: The paper offers a good contribution, but I have the following modifications:

1. The "Abstract and Conclusion" sections need improvement; authors should rewrite these sections.

2. There are a few grammatical errors in the paper. The authors need to review the full text carefully.

3. There are several recently published papers that are relevant to this manuscript, so some or all of them should be cited, such as 1. https://doi.org/10.1007/s40520-023-02483-y

2. doi: https://doi.org/10.1080/07853890.2023.2275148

3. doi: https://doi.org/10.1111/jpm.12990

4. doi: 10.1186/s12960-019-0440-y

5. doi: 10.1057/s41599-023-01719-6

6. doi: https://doi.org/10.1063/5.0156458

7. doi: https://doi.org/10.1016/j.ymssp.2022.110022

4. In some equations, some symbols are not defined. Please correct this.

5. Add more statistical details about the linear regression model.

6. Adding future work and the limitations of this research.

7. add the dataset file.

=== == === ==== === === == === ==== = = == = ==

Reviewer #7: I am pleased to review the manuscript title "Common misconceptions held by health researchers when interpreting linear regression assumptions, a cross-sectional study". The area is interesting however the manuscript is not suitable for publication. Firstly, the concept of common misconceptions are unclear. Secondly, it is not clear what methodology was adopted for randomness to select the sample. Thirdly, the response variable, an article allocation is purely subjective and there is no uniform policy adopted.

Reviewer #8: Summary of the Manuscript

The authors analyzed 95 papers published in PLOS ONE in 2019 that utilized linear regression. The results reveal that only 37% of authors checked any linear regression assumptions, and none reported checking all assumptions. A key finding is the widespread misconception that the dependent variable (Y) should be tested for normality instead of the residuals, with only 5 papers correctly addressing residual normality. The study underscores a significant gap in statistical reporting practices and provides actionable recommendations for improving these practices.

Strengths

Relevance: The paper addresses a critical issue in statistical reporting, which directly impacts the reliability of health research findings.

Robust Methodology: The random sampling of papers, dual-statistician reviews, and use of Gwet’s statistic to measure agreement lend credibility to the study.

Actionable Insights: The recommendations for teaching, journal reviewing practices, and statistical training are clear and applicable across research domains.

Suggestions for Improvement

Clarity in Statistical Concepts: The paper could elaborate more on why residual-based checks are critical and provide more detailed explanations of statistical terms like homoscedasticity and independence to ensure accessibility for non-specialist readers.

Consistency in Terminology: Terms like "errors" and "residuals" are occasionally used interchangeably. Providing consistent definitions would enhance clarity.

Detailed Visual Examples: Including additional visuals, such as examples of incorrect assumption checks and problematic residual patterns, would strengthen the practical impact of the findings.

Broader Context: Comparing the findings with reporting practices in other disciplines, such as psychology or social sciences, could help contextualize whether the observed issues are unique to health research or widespread.

Strengthening Recommendations: The recommendations could benefit from suggesting specific tools or frameworks (e.g., automated statistical checks, STROBE guidelines) to help authors and reviewers improve adherence to best practices.

Discussion of Implications: Expand the discussion on how poor assumption checking impacts the validity of health research outcomes, potentially leading to ineffective or harmful clinical interventions.

While the authors address several limitations, such as the focus on a single journal and variability among statistical raters, they could include suggestions for addressing these issues in future research. Additionally, acknowledging potential improvements in the study design (e.g., inclusion of papers from multiple journals or disciplines) would add transparency.

This manuscript addresses a critical issue in statistical reporting and provides a foundation for improving research practices in health and biomedical sciences. I recommend accepting the paper after minor revisions to enhance clarity, accessibility, and contextual depth.

Best regards

Reviewer #9: In Linear Regression section:

lines 195 - 197:

"... represented as a straight line", this is not always the case, as it oversimplifies the form of linear regression. As has been pointed out, the "linear" term is not for the relationship between the Y and X variables, but between Y and the regression coefficients, which imply that the representation of relationship between X and Y (in the case of 1 predictor) can be of any form, e.g. it can be a bit curvature, for example if the relationship is Y=B0+B1log(X).

Probably it needs more explanation on why the assumption checking is required. In line 201; it is written that linear regression are most commonly fit using OLS. How do you know? Moreover, when using OLS, there is no requirement of the assumptions (unlike if using maximum likelihood estimator), so this might be the case that the majority of authors (of papers examined in this study) were not aware of the assumptions. I suggest mentioning about the Gauss-Markov theorem, that if the OLS estimates fulfill the assumptions, then the BLUE (Best Linear Unbiased Estimators) are obtained, and thus the generalization is allowed.

lines 213 - 214: It is written that the intercept is the estimated value of Y when X=0. While this is mathematically correct, but it is not statistically fully accepted, as this might be misleading, that is interpreting the intercept as the predicted Y-value when X=0, which is a serious pitfall in using regression. If the data does not contain X=0, then the interpretation of the intercept sometimes does not make sense. the intercept is "merely" an adjustment made when estimating the regression parameters so that they will fulfill the condition of SE (sum of error) = 0.

6. PLOS authors have the option to publish the peer review history of their article (what does this mean?). If published, this will include your full peer review and any attached files.

Reviewer #1: **Yes: **Dr. Asad Ali

Reviewer #2: No

Reviewer #3: No

Reviewer #4: No

Reviewer #5: No

Reviewer #6: No

Reviewer #7: No

Reviewer #8: No

Reviewer #9: **Yes: **Sarini Abdullah

---

## [Author Response · Author response to Decision Letter 1]

20 Feb 2025

My response to the editor and the nine reviewers can be seen in the resonse to reviewers document.

---

## [Decision Letter · Decision Letter 1]

Common misconceptions held by health researchers when interpreting linear regression assumptions, a cross-sectional study

PONE-D-24-05674R1

Dear Dr. Jones,

We’re pleased to inform you that your manuscript has been judged scientifically suitable for publication and will be formally accepted for publication once it meets all outstanding technical requirements.

Kind regards,

Mohamed R. Abonazel, Ph.D.

Academic Editor

PLOS ONE

Reviewers' comments:

Reviewer's Responses to Questions

**Comments to the Author**

1. If the authors have adequately addressed your comments raised in a previous round of review and you feel that this manuscript is now acceptable for publication, you may indicate that here to bypass the “Comments to the Author” section, enter your conflict of interest statement in the “Confidential to Editor” section, and submit your "Accept" recommendation.

Reviewer #3: All comments have been addressed

Reviewer #5: All comments have been addressed

Reviewer #6: All comments have been addressed

Reviewer #8: All comments have been addressed

2. Is the manuscript technically sound, and do the data support the conclusions?

Reviewer #3: Yes

Reviewer #5: (No Response)

Reviewer #6: Yes

Reviewer #8: Yes

3. Has the statistical analysis been performed appropriately and rigorously? 

Reviewer #3: Yes

Reviewer #5: (No Response)

Reviewer #6: Yes

Reviewer #8: Yes

4. Have the authors made all data underlying the findings in their manuscript fully available?

Reviewer #3: Yes

Reviewer #5: (No Response)

Reviewer #6: (No Response)

Reviewer #8: Yes

5. Is the manuscript presented in an intelligible fashion and written in standard English?

Reviewer #3: Yes

Reviewer #5: (No Response)

Reviewer #6: (No Response)

Reviewer #8: Yes

6. Review Comments to the Author

Reviewer #3: (No Response)

Reviewer #5: (No Response)

Reviewer #6: (No Response)

Reviewer #8: (No Response)

7. PLOS authors have the option to publish the peer review history of their article (what does this mean?). If published, this will include your full peer review and any attached files.

Reviewer #3: **Yes: **Wubishet Gezimu

Mattu University, Ethiopia

Reviewer #5: No

Reviewer #6: No

Reviewer #8: **Yes: **Nadeem Akhtar

---

## [Editor Report · Acceptance letter]

PONE-D-24-05674R1

PLOS ONE

Dear Dr. Jones,

I'm pleased to inform you that your manuscript has been deemed suitable for publication in PLOS ONE. Congratulations! Your manuscript is now being handed over to our production team.

Kind regards,

on behalf of

Dr Mohamed R. Abonazel

Academic Editor

PLOS ONE